# An AMPK-dependent, non-canonical p53 pathway plays a key role in adipocyte metabolic reprogramming

Hong Wang[1], Xueping Wan[1], Paul F Pilch[2], Leif W Ellisen[3,4], Susan K Fried[5], Libin Liu[1]*

[1]Departments of Pharmacology & Experimental Therapeutics, Boston University, School of Medicine, Boston, United States; [2]Biochemistry, Boston University, School of Medicine, Boston, United States; [3]Massachusetts General Hospital Cancer Center, Boston, United States; [4]Harvard Medical School, Boston, United States; [5]Diabetes Obesity and Metabolism Institute, Icahn School of Medicine at Mount Sinai, New York, United States

**Abstract** It has been known adipocytes increase p53 expression and activity in obesity, however, only canonical p53 functions (i.e. senescence and apoptosis) are attributed to inflammation-associated metabolic phenotypes. Whether or not p53 is directly involved in mature adipocyte metabolic regulation remains unclear. Here we show p53 protein expression can be up-regulated in adipocytes by nutrient starvation without activating cell senescence, apoptosis, or a death-related p53 canonical pathway. Inducing the loss of p53 in mature adipocytes significantly reprograms energy metabolism and this effect is primarily mediated through a AMP-activated protein kinase (AMPK) pathway and a novel downstream transcriptional target, lysosomal acid lipase (LAL). The pathophysiological relevance is further demonstrated in a conditional and adipocyte-specific p53 knockout mouse model. Overall, these data support a non-canonical p53 function in the regulation of adipocyte energy homeostasis and indicate that the dysregulation of this pathway may be involved in developing metabolic dysfunction in obesity.

*For correspondence:
libin@bu.edu

Competing interests: The authors declare that no competing interests exist.

## Introduction

White adipose tissue (WAT) plays a central role in nutrient homeostasis, serving as the site of calorie storage and the source of key adipokines (*Guilherme et al., 2008*; *Rosen and Spiegelman, 2006*; *Kahn et al., 2019*). Obesity is a condition of excess lipid accumulation in white adipose tissue and has been recognized as a major risk factor for many human diseases, including type II diabetes, cardiovascular disease, and some types of cancer. As the major cell type of fat tissue, white adipocytes store triglycerides, which are synthesized from glucose and fatty acids, and can be broken down through lipolysis when needed as fuel. Controlled by multiple fine-tuned regulatory mechanisms, adipocytes have the ability to sense and coordinate responses to changes in nutrient availability during fasting and feeding, and they play central roles in lipid and glucose metabolism. Dysregulations in these processes are the direct results of insulin resistance and occur at the early stages of metabolic syndrome. Understanding the underlying regulatory mechanisms is critical for developing new early diagnostic tools and treatments for obesity-related metabolic diseases.

Traditionally, p53 has been considered a tumor suppressor that can be activated by a variety of stress signals, such as DNA damage, oncogene activation, and other stresses (*Vousden and Prives, 2009a*). Once activated, the p53 pathway has a wide range of downstream effects, the most prominent of which are cell death, senescence, cell cycle arrest, and autophagy (*Kruiswijk et al., 2015*; *Maddocks and Vousden, 2011*). In addition, recent research has shown that p53 plays a significant

role in metabolic regulation by inhibiting aerobic glycolysis while promoting mitochondrial oxidative phosphorylation and that the disruption of p53 function leads to the 'Warburg effect,' which is an enhanced conversion of glucose to lactate in cancer cells (*Vousden and Ryan, 2009b*; *Berkers et al., 2013*; *Kung and Murphy, 2016*). Although most of these studies focused on the malignant transformation of cancer cells, it is clear that p53 activation is closely associated with general metabolic regulation as well (*Kung and Murphy, 2016*; *Krstic et al., 2018*). In addition, several studies have suggested that p53 is involved in normal hepatic glucose homeostasis and lipid metabolism in liver tissue (*Prokesch et al., 2017*; *Derdak et al., 2013*). Studies of skeletal muscle also indicate that p53 is important for muscle mitochondrial function (*Park et al., 2009*). The metabolic role played by p53 is usually associated with cell adaptive responses for survival, which is distinct from the cell death and senescence-related regulations observed in the genotoxic stress conditions. Different regulatory mechanisms for the activation of each pathway have been identified, including upstream signaling, post-translational modifications, and downstream activated target genes (*Vousden and Prives, 2009a*; *Kruiswijk et al., 2015*).

Studies of adipose tissue have observed increases in the expression and activity of p53 in the adipocytes of obese mice (*Yahagi et al., 2003*). Adipocyte-specific knockout mice (*Minamino et al., 2009*; *Shimizu et al., 2012*; *Shimizu et al., 2013*) have been generated to study the role of p53 in metabolism, but these studies had a number of limitations. (i) Indirect association: Data from whole-body p53 knockout mice studies are inconsistent and suggest that the observed metabolic changes may not be directly attributed to adipose tissue, but rather the cumulative effect of p53 deficiency in other cells and tissues in those mouse models. (ii) A focus only on canonical p53 function: Other studies have manipulated p53 in adipocytes but predominantly focused on adipocyte death and apoptosis (*Minamino et al., 2009*; *Shimizu et al., 2012*; *Shimizu et al., 2013*), observing that metabolic phenotypes were exclusively attributed to cell death-related tissue inflammation without addressing cellular energy flux. (iii) *Fabp4* promoter: In addition, all adipocyte-specific p53 studies have used a system in which p53 ablation was driven by the *Fabp4* (fatty-acid-binding protein 4, aP2) promoter (*Fabp4*-Cre mice crossed with p53 floxed mice). However, the aP2 promoter has been known to be potentially active in tissues other than adipose, such as macrophages (*Fu et al., 2000*; *Fu et al., 2002*; *Makowski et al., 2001*). These inadvertent modifications of p53 levels in tissues other than adipose tissue may distort the data, especially in studies of the mechanisms of adipose tissue inflammation or systemic nutrient challenges. (iv) Lastly, a lack of attention to adipocyte differentiation affected by p53: p53 is known to regulate adipocyte differentiation and fat tissue development in various ways (*Thompson et al., 1998*; *Inoue et al., 2008*; *Okita et al., 2014*) that may significantly impact energy metabolism when common germline or embryonic p53 knockout mouse models are used. Due to these limitations, further studies into the physiological role of p53 in adipocyte metabolism are needed.

To this end, we created inducible in vitro and *in-vivo* working models, in which adipocyte-specific p53 expression levels could be manipulated at precisely controlled time points, so that p53 could be expressed normally during adipocyte differentiation and tissue development. The results from these working systems demonstrated a central role played by p53 in adipocyte energy metabolic reprogramming under challenging metabolic conditions. This specific regulation is mediated through lysosomal acid lipase (LAL), a downstream transcriptional target of p53. Additionally, AMP-activated protein kinase (AMPK)-dependent phosphorylation is involved. Both factors differentiate p53 metabolic function from its traditional canonical pathway that are related to cell senescence, apoptosis, and death. Furthermore, this non-canonical p53 function explains the metabolic effect of metformin in adipocytes and has the potential to generate new pharmaceutical approaches for modulating the p53 pathway and, thus, directly improving adipocyte functionality.

## Results

### In adipocytes p53 can be up-regulated by nutrient starvation, but without canonical pathway activation

In healthy mice, we observed that adipocytes express the p53 protein under normal conditions and that p53 expression can be significantly up-regulated with 24 hr fasting and down-regulated with refeeding (*Figure 1A*). We also observed that a short-term (3 days) switch from a low-fat (Research

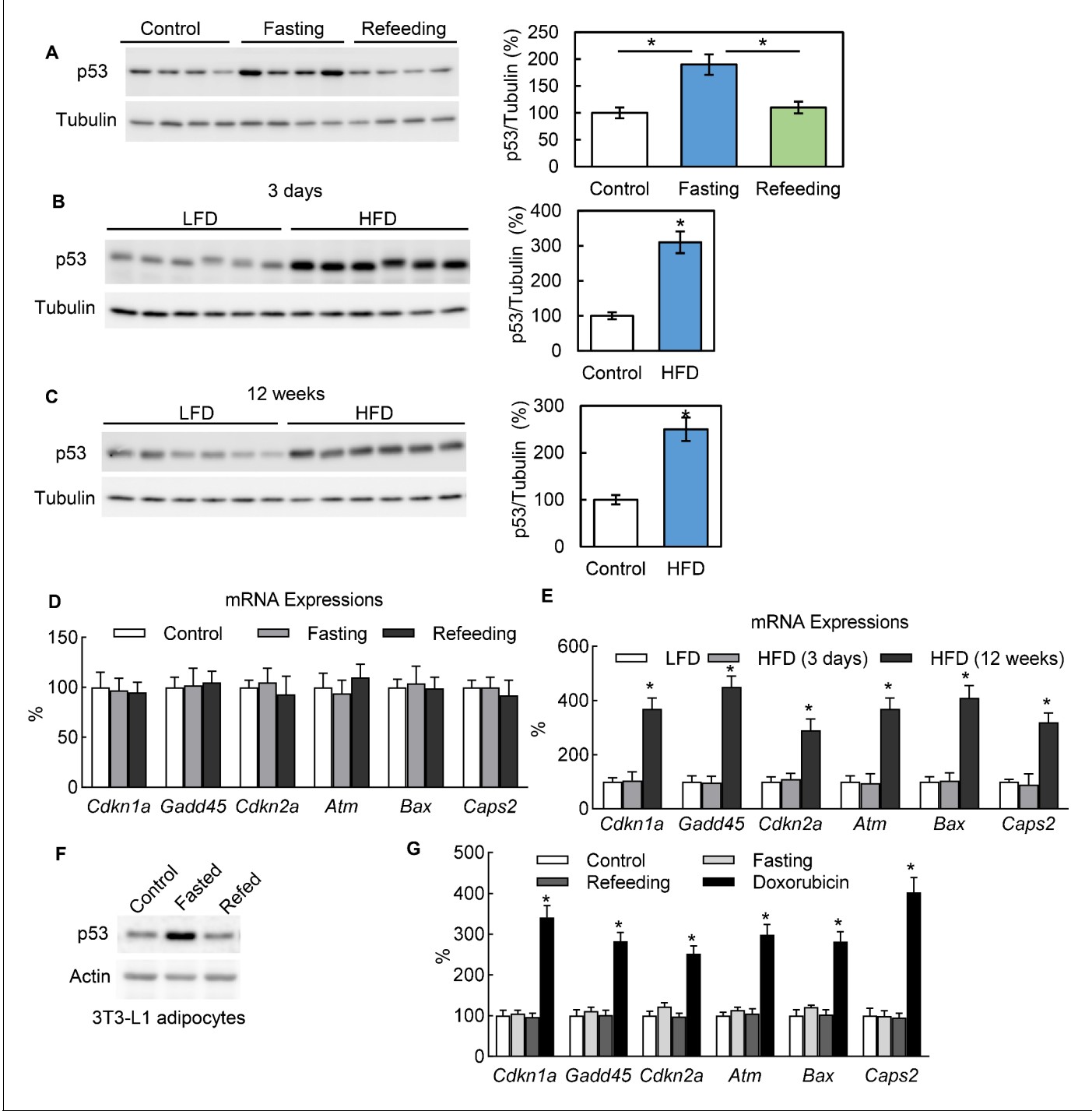

**Figure 1.** In adipocytes p53 can be up-regulated by nutrient starvation, but without canonical pathway activation. (**A**) Three groups of WT C57 mice were either not treated (Control), fasted for 24 hr (Fasting), or followed by refeeding for 24 hr (Refeeding). Primary adipocytes isolated from epididymal fat tissue were subjected to western blot using indicated antibodies. (**B–C**) p53 protein levels in isolated primary adipocytes from mice fed a high-fat diet (HFD) for 3 days (**B**) or 12 weeks (**C**) were examined by western blots. The quantification of relative p53 expression levels was normalized by tubulin and shown in bar graphs. (**D–E**) Gene expressions of p53 canonical pathway were measured by RT-PCR in samples from A-C. (**F**) The protein expression levels of p53 and actin were measured from 3T3-L1 adipocytes cultured in three media conditions: normal culture (DMEM with 10% FBS), fasting (PBS with 1% BSA and ISO, 10 mM for 6 hr), and refeeding (normal plus insulin for 6 hr after fasting). (**G**) Gene expressions of p53 canonical pathway were measured by RT-PCR in samples from F.

The online version of this article includes the following source data for figure 1:

**Source data 1.** An Excel sheet with numerical quantification data.

Diet, D1245B) to high-fat diet (HFD) (D12492) significantly up-regulated the p53 protein abundance in the adipocytes of C57BL/6 mice as well (*Figure 1B*). In addition, p53 can be up-regulated with long-term (12 wks) HFD feeding (*Figure 1C*). However, the p53 pathway downstream genes traditionally related to cell senescence, apoptosis, and death, such as *Cdkn1a*, *Gadd45*, and *Cdkn2a*, behaved differently. These markers were up-regulated by long-term HFD but unaffected by 24 hr fasting and short-term (3 days) HFD feeding (*Figure 1D–E*), which suggests the non-canonical function of p53 may be activated under these short-term metabolically challenging conditions. This phenomenon can be consistently observed in cultured 3T3-L1 adipocytes. To recreate the systemic effects of fasting in the cultured 3T3-L1 adipocytes of mice, we added isoproterenol (ISO, 10 μM) to a nutrient starvation medium (PBS with 1% BSA). As shown in *Figure 1F*, after switching from a normal culture medium (10% FBS in high-glucose DMEM) to the fasting medium for 6 hr, the p53 expression level was significantly up-regulated. However, refeeding (with a normal culture medium with 10 nM insulin) reduced the p53 expression back to a non-fasting condition. Of note, this p53 up-regulation was not associated with canonical pathway activation in contrast to doxorubicin, a DNA replication inhibitor used widely as a p53 pathway activator for cell senescence, cell cycle arrest, and cell death (*Figure 1G*). Since fasting can cause adipocytes to undergo significant reprogramming of cellular energy metabolism, these data suggest that p53 may be directly involved in adipocyte metabolic regulation.

## Deletion of p53 increases adipocyte glucose metabolism

Previous research has shown that p53 affects adipocyte differentiation, which could have profound and indirect effects on adipocyte metabolism. To focus on mature adipocytes and avoid these effects on adipocyte differentiation, we generated conditional CRISPR-Cas9 genome-edited p53 knockout 3T3-L1 stable cell lines (p53–KO) using an rtTA-TRE inducible lentivirus system. In this working model, *Trp53* was deleted after 8 days' differentiation, when the 3T3-L1 adipocytes were fully differentiated. As shown in *Figure 2A*, p53 can be effectively eliminated after a 3 day doxycycline treatment. To protect against any unexpected effects from doxycycline, all wild-type (WT) controls were treated with doxycycline in the same way as the KO cells. Using this working model, we first examined the role of p53 in glucose metabolism. After the p53-KO and WT 3T3-L1 adipocytes were incubated overnight with Krebs-Ringer modified buffer and 1% BSA, the culture mediums were switched with 5 mM glucose and 0.1% FBS in no-glucose DMEM, and the changes in glucose concentrations for the culture media were monitored for 60 hr. As shown in *Figure 2B*, glucose concentration in the culture medium of p53-KO adipocytes decreased much more quickly than WT control cells, indicating that the deletion of p53 can enhance adipocyte glucose consumption. Consistent with these results, a higher glycolytic capacity and glycolytic reserve were observed in the p53-KO 3T3-L1 adipocytes, as indicated by their extracellular acidification rates, which were measured using Seahorse XF technology (*Figure 2C–D*). When we measured the basal and insulin-stimulated glucose uptake, the differences were not statistically significant (data not shown) although the p53-KO adipocytes tended to have higher rates, and the expressions of GLUT1 and GLUT4 in the adipocytes were unaffected by the deletion of p53 (data not shown), which suggested that the p53 reduced the glucose metabolism, though not through a glucose transport-dependent mechanism. The pentose phosphate pathway (PPP) is a metabolic pathway that branches off from glycolysis and generates metabolites that can synthesize nicotinamide adenine dinucleotide phosphate hydrogen (NADPH). As shown in *Figure 2E*, p53 deletion led to an increase in NADPH levels. However, as shown in *Figure 2F*, these findings did not result from gene expression changes in glucose-6-phosphate dehydrogenase (G6PD) (*Jiang et al., 2011*). Additionally, glycolysis produces the intermediate metabolite pyruvate. Under anaerobic conditions, pyruvate is converted to lactic acid by lactate dehydrogenase (LDH) (*Hagen and Ball, 1960*; *Schmidt and Katz, 1969*; *Newby et al., 1988*; *Frayn and Coppack, 1990*; *DiGirolamo et al., 1992*). In the present study, the lactate production of the p53-KO adipocytes was significantly increased, comparing to WT control cells (*Figure 2G*). However, the gene expression levels of the *Ldha* and lactate transporters MCT1 (*Slc16a1*) and MCT4 (*Slc16a4*) (*Ross et al., 2010*; *Halestrap, 2013*) were not affected (data not shown). Overall, these data demonstrate that although inhibiting p53 activated the adipocyte glucose metabolic pathways, the enzymes/factors directly related to these pathways were not affected.

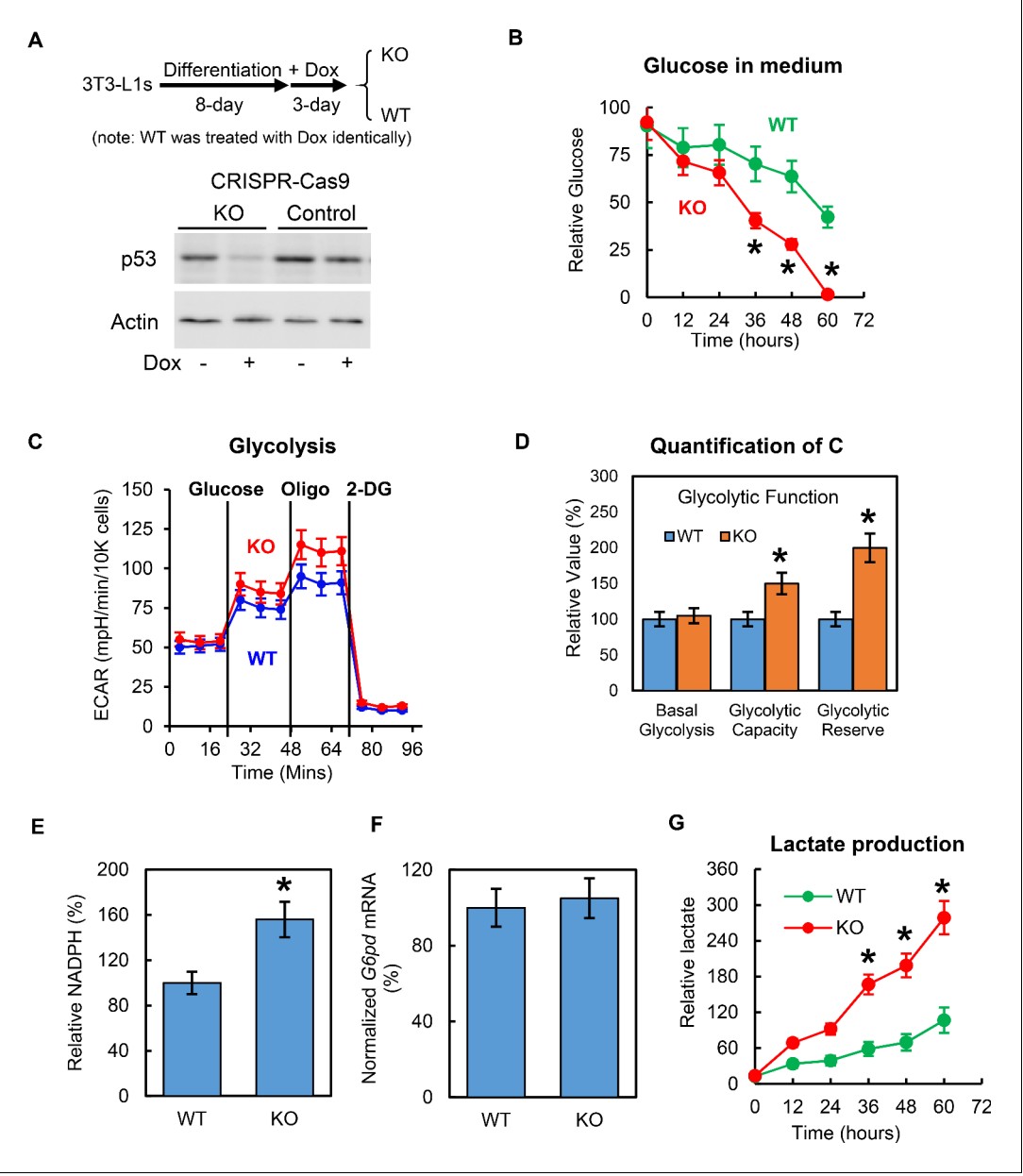

**Figure 2.** Deletion of p53 increases adipocyte glucose metabolism. (**A**) In vitro working model for inducible p53 knockout (KO) in 3T3-L1 adipocytes. (**B**) p53-knockout (KO) and wild-type (WT) cells were cultured in no-glucose DMEM with 0.1% FBS for overnight, then glucose was added to media to a final concentration of 5 mM. 10–20 μl culture media were sampled every 12 hr. Glucose concentrations were measured using commercially available kits. (**C–D**) Glycolytic and mitochondrial functions were measured by Seahorse. Relative NADPH levels (**E**) and G6DP mRNA expression levels (**F**) were examined. (**G**) Lactate levels were measured similarly as in (**B**). (*$p < 0.05$).
The online version of this article includes the following source data for figure 2:

**Source data 1.** An Excel sheet with numerical quantification data.

## Deletion of p53 suppresses adipocyte mitochondrial respiration and fatty acid oxidation

To further understand how p53 mediates metabolic changes in adipocytes, we examined whether p53 affected adipocyte mitochondrial function. The oxygen consumption rates (OCRs) for the p53 knockout and WT cells were measured using Seahorse Flux Analyzer. Both the basal and maximum respiration rates were lower in the p53-KO adipocytes than in the WT controls (*Figure 3A*),

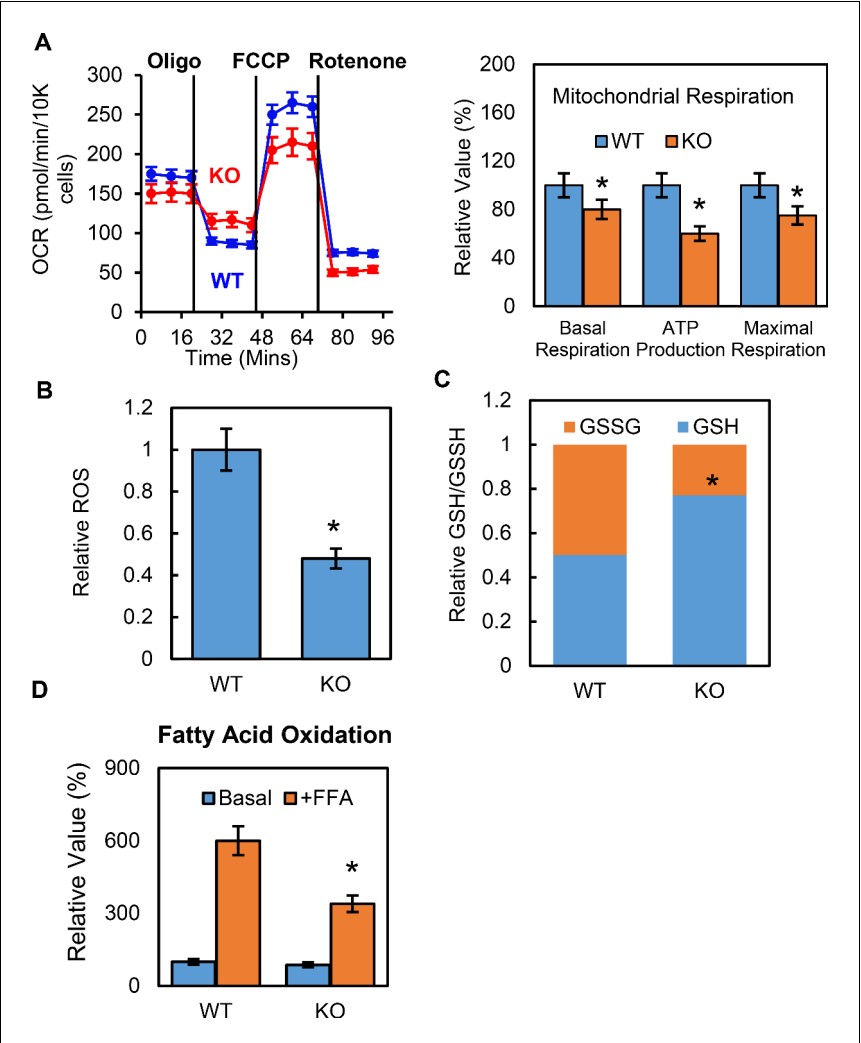

**Figure 3.** Deletion of p53 suppresses adipocyte mitochondrial respiration and fatty acid oxidation. (**A**) Mitochondrial respiration was measured by Seahorse XF technology in p53-KO and WT control 3T3-L1 adipocytes. (**B–C**) ROS, GSSG, and GSH were measured using commercially available kits. (**D**) Fatty acid oxidation was measured using Seahorse XF technology. (*$p < 0.05$).

The online version of this article includes the following source data for figure 3:

**Source data 1.** An Excel sheet with numerical quantification data.

suggesting that mitochondrial function was suppressed. Reactive oxygen species (ROS) are constantly produced during normal metabolism, especially through oxidative phosphorylation. When we used Amplex Red to measure the basal cellular ROS levels in the p53-KO adipocytes in vitro, we found that the p53 knockout adipocytes showed lower ROS levels when compared to the WT cells (*Figure 3B*). We also measured the GSH and GSSG levels and their ratios and calculated the relative redox potentials. Compared to the WT cells, the p53-KO adipocytes had much higher GSH levels and redox potential (*Figure 3C*). These data suggest that p53 plays a key role in promoting mitochondrial respiration and ROS generation in adipocytes and that deletion of p53 could have the opposite effect. However, this suppression was not due to changes in the amount of mitochondria, since the copy number of mitochondria was not affected by the p53 (data not shown).

Glucose and fatty acids are two main energy sources for mitochondrial respiration, but p53-KO cells have a high glycolysis capacity, which led us to ask if fatty acid oxidation was regulated by p53 in adipocytes. Using the Seahorse XF palmitate-BSA FAO substrate, we performed fatty acid oxidation (FAO) assays in the p53-iKO and WT 3T3-L1 adipocytes. As shown in *Figure 3D*, the OCR-to-

basal fold change increase was much lower in the p53-KO adipocytes, indicating reduced levels of fatty acid oxidation, which suggested that fewer free fatty acids were available as energy fuel in the p53-KO adipocytes.

## Deletion of p53 suppresses adipocyte lipid metabolism through the direct down-regulation of LAL

To investigate the role of p53 in adipocyte lipid metabolism, we measured the lipolysis activity and fatty acid oxidation. The p53 knockout and WT 3T3-L1 adipocytes were incubated overnight with a fresh full medium (10% FBS, high-glucose DMEM), then switched to a no-glucose or no-lipid medium (PBS with 10 μM ISO and 0.5% BSA), after which the free fatty acid concentrations in the medium were monitored. Although the acute (30 min) isoproterenol stimulated lipolysis rates in the p53 knockout cells were not affected (data not shown) when compared to the WT cells, the long-term (>12 hr) lipolysis rates were significantly decreased (*Figure 4A*). This change was not due to the traditional neutral lipolysis pathway, since the total protein expression and phosphorylated levels of perilipin and HSL were the same between the p53 knockout and WT cells (*Figure 4—figure supplement 1*).

In recent years, the discovery of lipophagy suggests that acid lipolysis is also involved in the degradation and turnover of lipid droplets (*Singh et al., 2009a*), and is mediated through LAL, the enzyme that is responsible for the breakdown of triglycerides and cholesteryl esters in lysosomes at pH 4.5–5 (33). Since previous studies have suggested lipophagy plays a role in adipocyte lipolysis (*Lettieri Barbato et al., 2013*; *Lizaso et al., 2013*), we investigated whether LAL-dependent acid lipolysis plays any role in mediating the effects of p53 in adipocyte lipolysis. As shown in *Figure 4B*, consistent with previous report (*Lettieri Barbato et al., 2013*), we found LAL gene expression level was up-regulated in 'fasted' (ISO in PBS with 1% BSA) 3T3-L1 adipocyte, and this effect was at least partially dependent on p53. Furthermore, we also found that LAL-RNAi knockdown 3T3-L1 adipocytes impaired ISO-stimulated lipolysis (*Figure 4C–D*), suggesting that acid lipolysis was activated after long-term ISO stimulation. In comparison, ATGL (Adipose triglyceride lipase, the enzyme catalyzes the first reaction of neutral lipolysis) RNAi knockdown showed significant downregulation of total lipolysis, and double RNAi knockdowns (LAL and ATGL) further downregulated lipolysis (Fig, 4C-D). These data support LAL-mediated and neutral lipolysis are the main pathways, however, we can't rule out the existence of other lipase that might be involved. The overexpression of LAL in p53 null adipocytes completely restored the down-regulation of lipolysis in the p53-KO adipocytes, suggesting that LAL-dependent acid lipolysis is involved (*Figure 4E–F*). Overall, these results indicate that p53-regulated LAL expression is directly involved in lipolysis during nutrient starvation.

## LAL is a transcriptional target of p53

It's been known typically, p53 binds to the target genes as a tetramer, which comprises two dimers that each binds a decameric half-site with the consensus sequence (p53 response-element, p53-RE, RRRCWWGYYY, R = A/G, W = A/T, Y = C/T) (*El-Deiry et al., 1992*; *Figure 5A*). When we examined the sequence of LAL promoter region, a highly p53-RE like sequence was found in both mouse and human. By using Chip-qPCR assay we demonstrated p53 bound to this region in fasted mouse adipocytes (*Figure 5B*). Furthermore, by using luciferase reporter assay we show p53-RE deleted construct (Δp53RE) had diminished transcriptional activities even when p53 was over-expressed (*Figure 5C–D*), supporting that p53 directly regulates LAL gene expression through transcription.

## Metabolic regulation of p53 is mediated through AMPK-dependent phosphorylation

Previous studies in non-adipocytes have suggested that glucose starvation causes p53 up-regulation associated with p53 phosphorylation through AMPK-dependent pathways (*Jones et al., 2005*; *Feng et al., 2005*; *Feng et al., 2007*; *Assaily et al., 2011*; *Armata et al., 2010*). Therefore, we examined if a similar mechanism existed in adipocytes. We found that p53 was serine/threonine (p-S/T) phosphorylated by nutrient starvation in 3T3-L1 adipocytes and that refeeding reversed this change (*Figure 6A*). When the cells were treated with the cell-permeable AMPK activator AICAR, the phosphorylation levels were significantly up-regulated (*Figure 6B–C*), whereas using compound C, an AMPK inhibitor (*Zhou et al., 2001*), before fasting significantly prevented fasting-induced p53

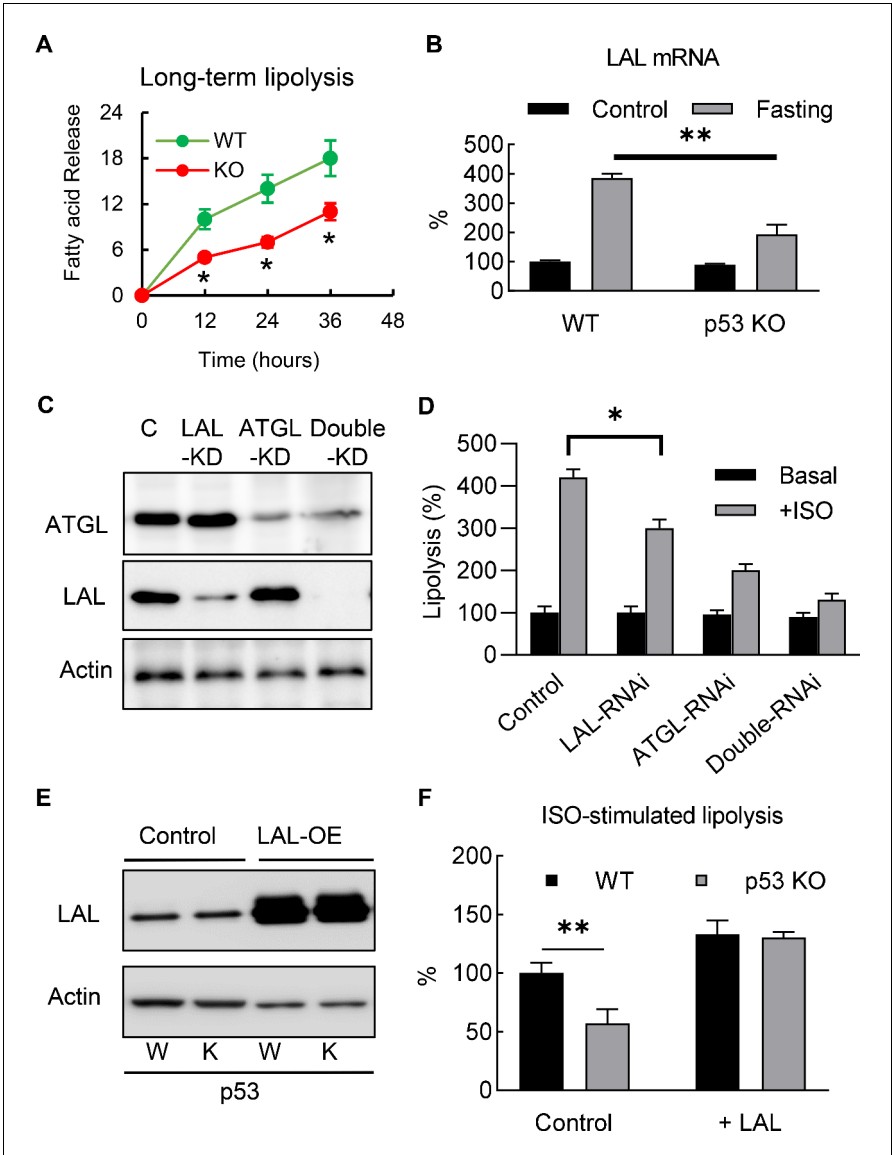

**Figure 4.** Deletion of p53 suppresses adipocyte lipid metabolism through the down-regulation of lysosomal acid lipase. (A) Long-term lipolysis activities were measured in Iso-stimulated p53-knockout (KO) and wild-type (WT) control adipocytes. (B) LAL mRNA expression levels were examined in the samples from the end time point (36 hr) of A (C) Protein expression levels of LAL, ATGL, and actin were examined by western blots in control, LAL, ATGL, and LAL/ATGL double RNAi knockdown cells. (D) 3T3-L1 adipocytes from C were cultured with or without ISO stimulation (in PBS +1% BSA) for 18 hr and the concentrations of fatty acid in the medium were measured. (*p<0.05). (E) LAL was overexpressed in WT and p53–KO 3T3-L1 adipocytes, and (F) long-term lipolysis (18 hr) activities were measured by determining FFA levels. (*p<0.05, **p<0.01).

The online version of this article includes the following source data and figure supplement(s) for figure 4:

**Source data 1.** An Excel sheet with numerical quantification data.

**Figure supplement 1.** Deletion of p53 does not affect traditional neutral lipolysis signaling pathway.

phosphorylation (*Figure 6B–C*). The regulation can be further demonstrated by examining the gene expression of p53 downstream target, LAL. As shown in *Figure 6D*, fasting-induced LAL gene up-regulation can be significantly inhibited by AMPK inhibitor, compound C.

Metformin is a widely used anti-diabetic drug that has clear benefits in relation to glucose metabolism and diabetes-related complications (*Sivalingam et al., 2014*; *Foretz et al., 2014*), but the mechanisms underlying these pleiotropic benefits are complex and still not fully understood.

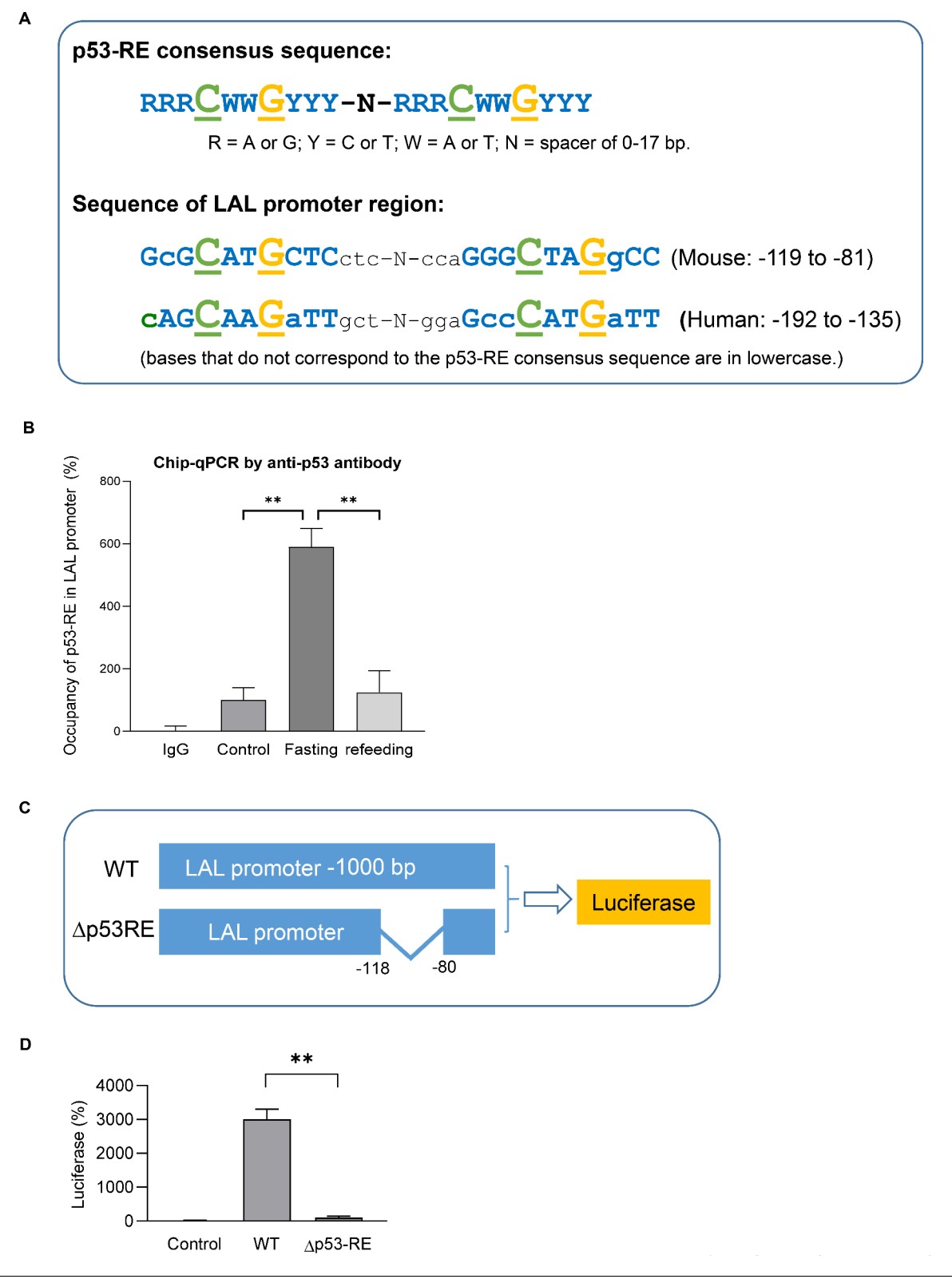

**Figure 5.** LAL is a transcription target of p53. (**A**) Previous known p53 response-element (p53–RE) consists of two copies of the palindromic half-site RRRCWWGYYY where each p53 monomer binds five nucleotides (top). A highly p53RE-like sequence was found in LAL promoter region in both human and mouse (bottom). (**B**) Chip-qPCR assay using IgG or p53 antibody from control, 24 hr fasted, and refed mouse primary adipocytes. (**C**) Luciferase reporter constructs were generated by using 1 kb wild-type (WT) or p53-RE deleted (Δp53RE) LAL promoter sequence. (**D**) 3T3-L1 adipocyte were

*Figure 5 continued on next page*

*Figure 5 continued*

transiently transfected by electroporation with constructs from C together with pcDNA3.1-p53 construct. After 48 hr, cells were harvested in the reporter lysis buffer. Luciferase activity in cell lysates was assayed as described under 'Experimental Procedures' and normalized by protein concentrations. Data are presented for triplicate samples as mean ± S.D. (error bars). **, p<0.01.

The online version of this article includes the following source data for figure 5:

**Source data 1.** An Excel sheet with numerical quantification data.

Findings also vary greatly depending on dosing and duration of treatment, with clear differences between acute and chronic administration (*Foretz et al., 2014*). Metformin is believed to act mainly through reducing hepatic glucose production by AMPK activation (*Zhou et al., 2001*; *Hundal et al., 2000*; *Shaw, 2005*). However, this mechanism has been challenged by studies in AMPK-deficient

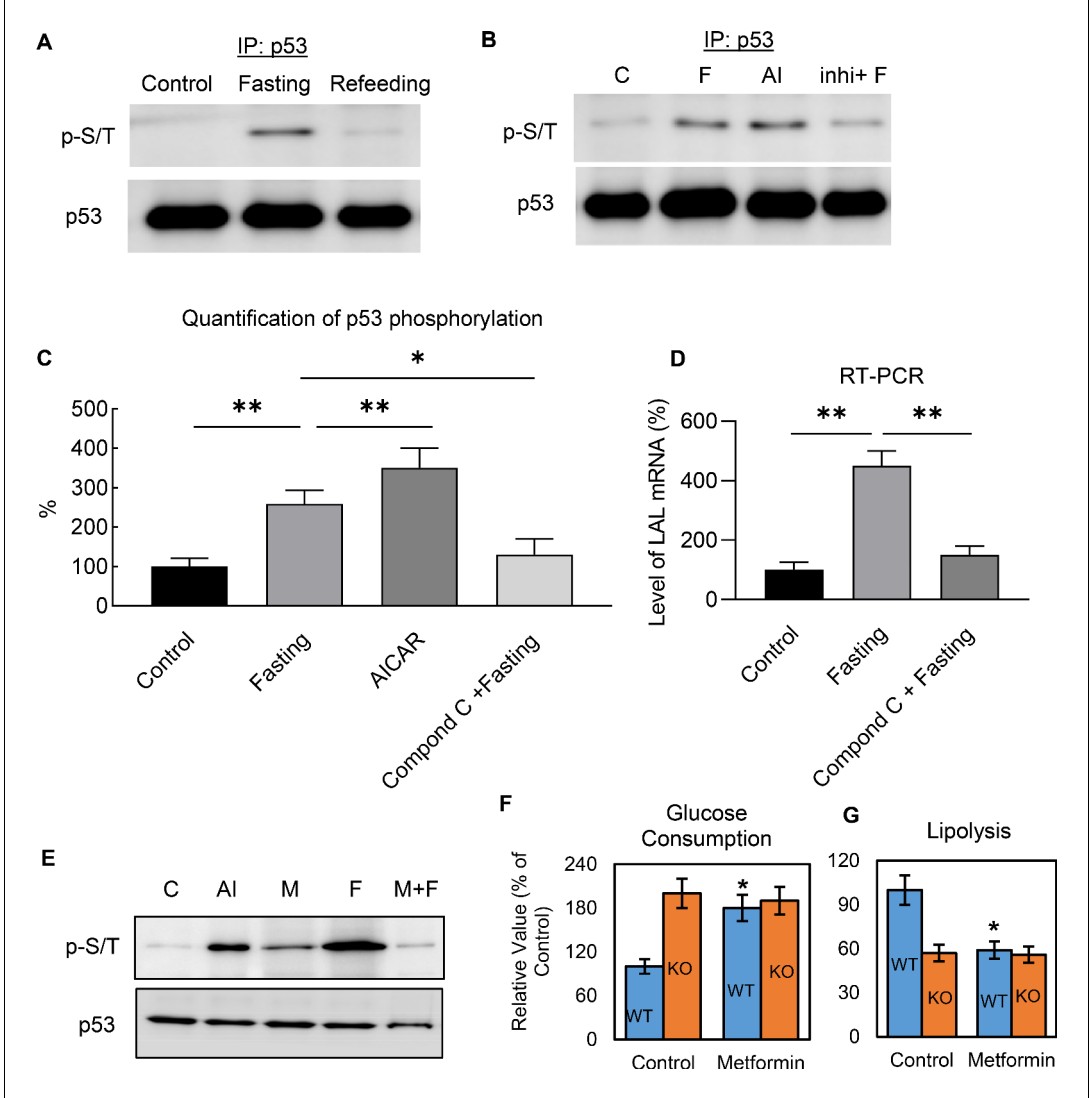

**Figure 6.** Metabolic regulation of p53 is mediated through AMPK-dependent phosphorylation. (**A–B**) Nuclear fraction was prepared from control (C, High-glucose DMEM with 10% FBS), fasted (F, PBS with 1% BSA and ISO for 6 hr), AICAR (AI), or compound C treated before fasting (Inhi+F), 3T3-L1 adipocytes, following by immunoprecipitation and western blots. (**C**) Quantification of B. (**D**) LAL mRNA expression levels were examined by RT-PCR in the cells from B. (**E**) The effect of metformin on fasting-induced p53 phosphorylation was determined by immunoprecipitation and western blots. (**F–G**) Effects of metformin on glucose consumption and lipolysis in cultured 3T3-L1 WT and p53-KO adipocytes were determined. (*p<0.05).

The online version of this article includes the following source data for figure 6:

**Source data 1.** An Excel sheet with numerical quantification data.

genetic mouse models (*Foretz et al., 2010*), in which metformin effects were still preserved, suggesting an AMPK-independent mechanism. Our data show although metformin can slightly induce p53 phosphorylation, the effect of 'fasting' was largely impaired (*Figure 6E*). Metformin treatment can increase glycolysis, and inhibit lipolysis, however, these effects were not observed in the p53 knockout 3T3-L1 adipocytes (*Figure 6F–G*), suggesting that metformin acts in a p53-dependent manner. These data not only further demonstrate that non-canonical p53 pathways in metabolic regulation, but also provide a new explanation for the metabolic effect of metformin in adipocytes.

### Inducible adipocyte-specific p53 knockout (p53-iAKO) mice show decreased lipolysis after fasting and improved metabolic phenotypes upon high-fat diet feeding

To study the physiological relevance of p53 in adipocytes, we generated a conditional adipocyte-specific p53 knockout mouse model (p53-iAKO) (*Figure 7A*). One of key mouse lines, adiponectin promoter-driven rtTA (inducible Tet-on system) was a gift from Dr. Phillip Scherer (*Sun et al., 2012*; *Wang et al., 2013*). In this model, when the mice were fed a diet containing doxycycline, the p53 expression in white adipose tissue was significantly reduced, specifically in comparison to the liver tissue (*Figure 7B–C*). With this model, we first examined the phenotypes under normal physiological conditions using the strategy shown in *Figure 7D*. In these baseline studies, we did not observe any obvious metabolic phenotypes, including body weight, overnight fasting glucose, and insulin, between the WT control and p53-KO mice (data not shown). However, when we challenged the mice with prolonged fasting (24 hr, started from 5 p.m.), we observed that the body and fat tissue weight losses, fasting glucose and serum-free fatty acid levels, and liver TG content were much lower in the p53-iAKO mice (*Figure 7E–I*), suggesting a potentially attenuated lipolysis ability within adipose tissue. Serum lactate concentration was slightly higher in p53-iAKO mice, but did not reach statistical significance, and blood pH remained unchanged (*Figure 7J–K*). Lactate tolerance test showed p53-iAKO mice had higher clearance rate for lactate, indicating up-regulated hepatic gluconeogenesis (*Figure 7L*). However, glucose uptakes were much higher in fat and muscle tissues, supporting increased glucose utilization in these tissues (*Figure 7M*), which probably explains the hypoglycemia observed in fasted p53-iAKO mice. Furthermore, the histology studies showed fat tissue from fasted p53-iAKO mice had less morphological change comparing to WT control mice (*Figure 7N*). Metabolic cage studies showed p53-iAKO mice had higher RER rate under fasting condition, suggesting they utilized more glucose as energy resource (Fig. O-P). Additionally, p53 and AMPK phosphorylation can be observed in fasted mice as well (*Figure 7Q*). Overall these data are consistent with our observations from the in vitro studies.

Previous studies of obesity have suggested that canonical p53 activations are associated with adipocyte senescence, apoptosis, and death (*Minamino et al., 2009*; *Shimizu et al., 2012*; *Shimizu et al., 2013*), which may explain the adipose tissue inflammation and metabolic dysfunction observed in obesity. However, these studies were conducted in long-term, diet-induced obesity (DIO) mice models. Our studies of cell culture systems led us to hypothesize that disturbances in non-canonical p53 metabolic regulations in the early stages of HFD feeding may directly contribute to metabolic dysregulation and the pathological development of insulin resistance. To test this theory, we performed a short-term HFD feeding study (60% calories from fat for six wks) (*Figure 7R*) and confirmed that the gene expressions related to inflammation and cell death were not up-regulated in this timeframe (data not shown). Under this condition we observed that the p53-iAKO mice had significantly improved metabolic phenotypes, including fasting glucose, insulin, fatty acid, and glucose tolerance test (IPGTT) (*Figure 7S–W*). These *in-vivo* data strongly indicate that adipocyte non-canonical p53 metabolism-specific functions play a crucial role in obesity-associated metabolic dysregulation.

## Discussion

In this report, we demonstrated that p53 plays a critical role in reprogramming adipocyte glucose and lipid metabolisms and that the down-regulation of p53 increases glycolysis capacity and inhibits lipolysis activity simultaneously, which may result in Warburg-like metabolic effects in adipocytes and contribute to increased adipocyte metabolic flexibility and insulin sensitivity in obesity. These findings are further supported by *in-vivo* studies using inducible adipocyte-specific p53 knockout mouse

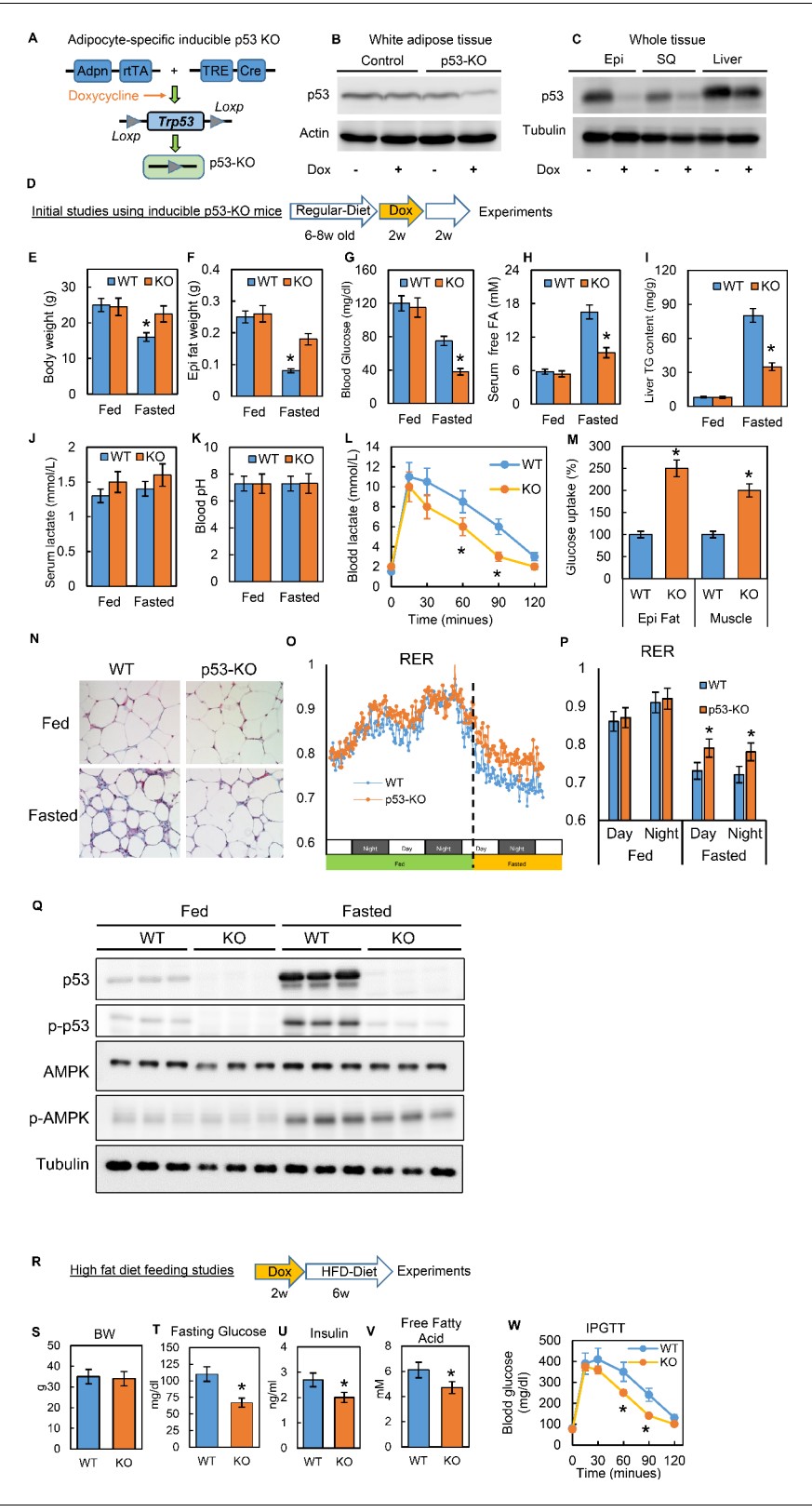

**Figure 7.** Inducible adipocyte-specific p53 knockout (p53-iAKO) mice show decreased lipolysis after fasting and improved metabolic phenotypes upon high-fat diet feeding. (**A**) iAdipo-p53-KO (KO) mouse was generated by breeding doxycycline-inducible (TRE-Cre), adipocyte-specific (Adpn-rtTA), floxed-p53 lines together, and induced by doxycycline. (**B**) white adipose tissue and (**C**) whole tissue from control (rtTA/Cre only) and KO mice fed with or without doxycycline diet were subjected to western blot by using indicated antibodies. (Epi: epididymal fat; SQ: subcutaneous fat). (**D**) Initial study

*Figure 7 continued on next page*

*Figure 7 continued*

design. (**E–K**) KO and WT mice were fasted for 24 hr and refed for 24 hr or not; body weight (**E**), epididymal fat tissue weight (**F**), blood glucose(**G**), serum-free fatty acid (**H**), liver TG content (**I**), serum lactate concentration (**J**), and blood pH (**K**) were measured. Lactate tolerance test (**L**) and in-vivo glucose uptake assay (**M**) were performed in WT and KO mice. (**N**) The histology H-E staining was conducted in epididymal fat tissues from WT and p53-KO mice under feeding (fed) and fasting (Fasted, 24 hr) conditions. (**O–P**) Respiratory exchange ratio (RER) was measured in metabolic cage studies. (**Q**) Total and phosphor-p53, AMPK, and tubulin expression levels in fed and fasted p53-knockout (KO) and wild-type (WT) mice were examined by western blots. (**R**) Study design of high-fat diet feeding studies. Metabolic phenotyping in high-fat diet-fed p53-knockout (KO) mice compared to WT, including (**S**) body weight (BW), (**T**) fasting glucose, (**U**) insulin, (**V**) fatty acids, and (**W**) IPGTT. (n = 3–6, *p<0.05).

The online version of this article includes the following source data for figure 7:

**Source data 1.** An Excel sheet with numerical quantification data.

models with HFD feeding. In general, 'metabolic flexibility' defines the ability of shifting energy source reliance between lipids and glucose in the whole-body or skeletal muscle in response to fasting or feeding (*Goodpaster and Sparks, 2017*). A similar concept can be applied when defining the metabolic functional capacities of adipocytes during continuous transitions between feeding and fasting cycles. Thus, adipocyte metabolic flexibility can be similarly determined by glucose uptake/glycolysis during feeding and lipolysis/lipid oxidation during fasting (*Figure 8*). In a healthy state, adipocytes will select either glucose or fatty acids as the main energy source in feeding and fasting conditions, continuously and cyclically switching between the two substrates. This transition is crucial for metabolic homeostasis and is achieved through multiple regulatory mechanisms for controlling the energy fuel flux. Our studies suggest that a non-canonical p53 pathway plays a key role in the regulation of adipocyte metabolic flexibility. This finding is distinct from previous studies, which concentrated on mediating genotoxic stress responses related to cell death or senescence. These results also suggest that adipocyte dysfunction in obesity may be caused by prolonged activation of this non-canonical p53 pathway. Therefore, p53-suppression-based approaches that beneficially

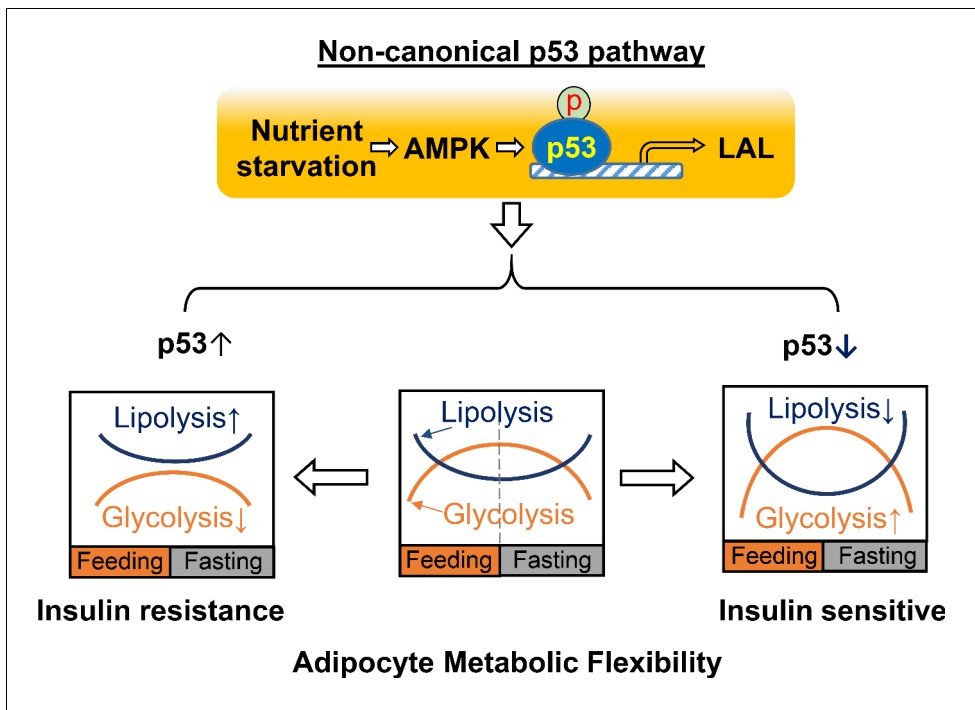

**Figure 8.** A working model for non-canonical p53 pathway-regulated adipocyte metabolic flexibility. Nutrient starvation causes an AMPK-dependent p53 non-canonical pathway activation, which leads to the upregulation of LAL-mediated lysosome acid lipolysis. This pathway plays key role for adipocyte oscillating between glycolysis and lipolysis under feeding and fasting states. In insulin resistance, p53 is found to be upregulated, while p53 downregulation is associated with insulin sensitivity.

reprogram adipocyte metabolic homeostasis in obesity may provide a new strategy for treating obesity and other metabolic diseases, including type II diabetes.

In attempting to understand the underlying mechanism of this non-canonical p53 pathway in metabolic regulation, we did not observe any significant changes to the key transporters or enzymes in glucose or lipid metabolic pathways, nor to any previously identified p53 downstream targets that were reportedly linked to the metabolic pathway in cancer cells (data not shown). It is not a surprise that these known regulators, which were identified in cancer cell studies, do not play significant roles in adipocytes. Instead, we found that p53 metabolic effects are mediated through its newly identified transcriptional target, LAL, which is an enzyme responsible for the breakdown of triglycerides and cholesteryl esters in lysosomes at pH 4.5–5 (*Dubland and Francis, 2015*). It is known that lipophagy-mediated acid lipolysis is involved in the degradation and turnover of lipid droplets (*Singh et al., 2009a*), particularly in hepatocytes. Prior studies have considered the major lipophagy function in adipocytes to be pre-adipocyte differentiation (*Singh et al., 2009b*; *Zhang et al., 2009*); however, the role of lipophagy in lipid droplet turnover in mature adipocytes remains unclear. The majority of studies have examined adipocyte lipolysis in relation to cytosolic neutral lipolysis (*Schweiger et al., 2006*; *Frühbeck et al., 2014*). However, studies using knockout mouse models revealed residual lipolytic activity, independent of neutral lipolysis, and suggested that LAL may potentially be involved (*Haemmerle et al., 2006*). Further studies have shown that lipophagy can be induced by fasting (*Lettieri Barbato et al., 2013*) or ISO stimulation (*Lizaso et al., 2013*) during 3T3-L1 adipocyte lipolysis. In addition, significant micro-lipid droplet (mLD) formation can be observed in 3T3-L1 adipocytes when stimulated by ISO (*Marcinkiewicz et al., 2006*; *Londos et al., 1999*; *Brasaemle et al., 2004*; *Moore et al., 2005*). Similar mLDs can be observed in the adipocytes of mice when treated with CL316,243 (*Wang et al., 2016*), which provides a potential opportunity to engage lysosomes for mLD degradation. To investigate this idea, we generated LAL knockdown 3T3-L1 adipocytes and found that the cells had a decreased ISO-stimulated lipolysis in comparison to the WT controls (*Figure 6*). These data indicate that LAL is directly involved in overall lipolysis during ISO stimulation. Furthermore, previous studies of human subjects with different types and degrees of obesity have revealed a direct correlation between autophagic activity and insulin sensitivity (*Kovsan et al., 2011*). Collectively, these studies indicate that LAL-mediated lysosome acid lipolysis plays a potential role in adipocyte lipid metabolism. Future studies need to clarify how LAL mediates the changes in glucose metabolism and mitochondrial functions and whether other p53 downstream targets exist.

Previous studies of non-adipocytes have suggested that glucose starvation causes p53 up-regulation, which is associated with p53 phosphorylation through AMPK or ATM (ataxia telangiectasia mutated) kinase-dependent pathways (*Jones et al., 2005*; *Feng et al., 2005*; *Feng et al., 2007*; *Assaily et al., 2011*; *Armata et al., 2010*), and is mainly associated with p53 cell apoptosis. In the present study, we demonstrated AMPK is also necessary for this non-canonical p53 pathway. In this regard, metformin has been believed to reduce hepatic glucose production by AMPK activation (*Zhou et al., 2001*; *Hundal et al., 2000*; *Shaw, 2005*) and it is a widely used anti-diabetic drug with clear benefits for glucose metabolism and diabetes-related complications (*Sivalingam et al., 2014*; *Foretz et al., 2014*), but the mechanisms underlying these pleiotropic benefits are complex and still not fully understood. Research findings related to metformin also vary greatly depending on the dosing and duration of treatment, with clear differences between acute and chronic administration (*Foretz et al., 2014*). Our studies show metformin treatment in adipocytes can activate p53 phosphorylation to some degree, but lead to the inhibition of fasting effect (*Figure 6E*). This apparent discrepancy may be explained through a negative-feedback regulatory mechanism. Metformin treatment may cause a pre-condition of AMPK-p53 pathway inhibition, however, the molecular details remains to be determined in future studies. Nevertheless, our data support metformin acts through a p53-dependent mechanism to affect adipocyte glucose consumption and lactate production (*Figure 6E-F*). AMPK seems to be a necessary factor for this non-canonical pathway, as a lower ATP/AMP ratio under fasting conditions can trigger the activation of AMPK, leading to the phosphorylation and transcriptional activation of p53. However, this finding cannot fully explain through AMPK-dependent mechanism in other challenging metabolic conditions, such as high-fat diet feeding, when a non-canonical p53 pathway is activated without a significant change in the ATP/AMP ratio. It is entirely possible that other factors are involved, which might also explain the regulation of LAL

transcription observed in this study. More studies are needed to fully understand the mechanistic details of the process.

Lastly, previous research has established a tight association between obesity and cancer (*Deng et al., 2016*; *Zwezdaryk et al., 2018*). Obese people have an increased risk of developing cancer and a poorer prognosis (*Lauby-Secretan et al., 2016*; *Kyrgiou et al., 2017*). In addition, 50% of cancer patients suffer from cachexia, which is a severe metabolic 'wasting' syndrome (*Vegiopoulos et al., 2017*). Previous studies have reported that metformin has anti-cancer effects (*Jalving et al., 2010*). It will be interesting to further study the connection between non-canonical and canonical p53 pathways and determine whether one can lead to the activation of the other. Future studies about the effect of the tumor suppressor p53 on adipocyte cellular metabolism may help us better understand the links between obesity and cancer.

# Materials and methods

## Key resources table

| Reagent type (species) or resource | Designation | Source or reference | Identifiers | Additional information |
|---|---|---|---|---|
| Genetic reagent (*Mus. musculus*) | B6.Cg-Tg(tetO-cre) (TRE-Cre mouse) | Jackson Laboratory | JAX stock #006234; RRID:IMSR_JAX:006234 | |
| Genetic reagent (*Mus. musculus*) | *Adipoq*-rtTA mouse | *Sun et al., 2012*; *Wang et al., 2013* | | A kind gift from Dr. Scherer. |
| Genetic reagent (*Mus. musculus*) | B6.129P2-Trp53tm1Brn/J (floxed-*Trp53* mouse) | Jackson Laboratory | JAX stock #008462; RRID:IMSR_JAX:008462 | |
| other | Doxycycline diet | BioServ | #S3888 | |
| other | Control diet | BioServ | #S4207 | |
| other | 60% high-fat diet | Research Diets | D12492 | |
| Cell line (mouse) | 3T3-L1 cell line. (mouse) | Zenbio | Zenbio: SP-L1-F; RRID:CVCL_0123 | |
| Transfected construct (mouse) | Mouse Lentiviral *Trp53* sgRNA, glycerol stock | GE Healthcare Dharmacon | GSGM11839-246656039 | Target sequence: CTGTACGGCG GTCTCTCCCA |
| Transfected construct (mouse) | Mouse Lentiviral *Trp53* sgRNA, glycerol stock | GE Healthcare Dharmacon | GSGM11839-247039414 | Target sequence: CTCCAGAAG ATATCCTGGTA |
| Transfected construct (mouse) | Mouse Lentiviral *Trp53* sgRNA, glycerol stock | GE Healthcare Dharmacon | GSGM11839-246656045 | Target sequence: GTGATGGGAG CTAGCAGTTT |
| recombinant DNA reagent | Inducible Cas9 lentiviral vector plasmid DNA | GE Healthcare Dharmacon | CAS11229 | TRE3G dox-inducible promoter. |
| recombinant DNA reagent | MGC mouse *Lipa* cDNA plasmid DNA | GE Healthcare Dharmacon | MMM1013-202770272 | Mammalian Expression LAL by CMV promoter |
| Transfected construct (mouse) | siRNA: *Lipa* | Thermo Fisher Scientific | AM16708 | siRNA ID 75920 |
| Transfected construct (mouse) | siRNA: *Lipa* | Thermo Fisher Scientific | AM16708 | siRNA ID 157079 |
| Transfected construct (mouse) | siRNA: *Lipa* | Thermo Fisher Scientific | AM16708 | siRNA ID 75827 |

*Continued on next page*

*Continued*

| Reagent type (species) or resource | Designation | Source or reference | Identifiers | Additional information |
|---|---|---|---|---|
| Transfected construct (mouse) | siRNA: *Pnpla2* | Thermo Fisher Scientific | AM16708 | siRNA ID 183465 |
| Transfected construct (mouse) | siRNA: *Pnpla2* | Thermo Fisher Scientific | AM16708 | siRNA ID 183466 |
| Transfected construct (mouse) | siRNA: *Pnpla2* | Thermo Fisher Scientific | AM16708 | siRNA ID 79116 |
| antibody | anti-p53 (rabbit polyclonal) | Santa Cruz | Santa Cruz: sc-6243; RRID:AB_653753 | WB. 4-degree overnight incubation only. 1:1000 |
| antibody | anti-p53 (mouse monoclonal) | Cell signaling | Cell signaling: 2524; RRID:AB_331743 | WB. 1:1000 |
| antibody | anti-p53 (mouse monoclonal) | Abcam | Abcam: ab26; RRID:AB_303198 | IP. 1:100 |
| antibody | anti-p53 (mouse monoclonal) | Thermo Fisher | Thermo Fisher: AHO0112; RRID:AB_2536305 | IP.1:100 |
| antibody | anti-Tubulin (mouse monoclonal) | Sigma | Sigma:T4026; RRID:AB_477577 | 1:2000 |
| antibody | anti-Actin (rabbit polyclonal) | Cell signaling | Cell signaling: 4967; RRID:AB_330288 | 1:1000 |
| antibody | anti-ATGL (rabbit polyclonal) | Cell signaling | Cell signaling: #2138; RRID:AB_2167955 | 1:1000 |
| antibody | anti-LAL (rabbit polyclonal) | Abcam | Abcam: ab154356; RRID:AB_154356 | 1:5000 |
| antibody | anti-Perilipin (Guinea Pig polyclonal) | Fitzgerald | Fitzgerald: 20R-PP004; RRID:AB_1288416 | 1:5000 |
| antibody | anti- Phospho-(Ser/Thr) PKA Substrate (rabbit polyclonal) | Cell Signaling | Cell signaling: #9621; RRID:AB_330304 | 1:1000 |
| antibody | anti-HSL (rabbit polyclonal) | Cell Signaling | Cell signaling: #4107; RRID:AB_2296900 | 1:1000 |
| antibody | anti-p-HSL (Ser563) (rabbit polyclonal) | Cell Signaling | Cell signaling: #4139; RRID:AB_2135495 | 1:1000 |
| antibody | anti-AMPKα (rabbit polyclonal) | Cell Signaling | Cell signaling: #2532; RRID:AB_330331 | 1:1000 |
| antibody | Anti-phospho-AMPKα (Thr172) (rabbit polyclonal) | Cell Signaling | Cell signaling: #2535; RRID:AB_331250 | 1:1000 |
| sequence-based reagent | *Lipa*-mRNA-F | This paper | qPCR primers | TGTGACCGAGATAATCATGCG |
| sequence-based reagent | *Lipa*-mRNA-R | This paper | qPCR primers | GAAGATACACAACTGGTCTGGG |
| sequence-based reagent | *Lipa*-promoter-F | This paper | Chip-qPCR primers | AAGCTCTGGCTGGGCTTAGAG |

*Continued on next page*

*Continued*

| Reagent type (species) or resource | Designation | Source or reference | Identifiers | Additional information |
|---|---|---|---|---|
| sequence-based reagent | *Lipa*-promoter-R | This paper | Chip-qPCR primers | GCAGGCGAGC TTGGCCAACCT |

## Animals

### Inducible adipocyte-specific p53 knockout mice (p53-iAKO)

p53-iAKO mice were generated by breeding doxycycline-inducible (TRE-Cre) (JAX stock #006234), adipocyte-specific (*Adipoq*-rtTA, a kind gift from Dr. Scherer) (*Sun et al., 2012*; *Wang et al., 2013*), and floxed-*Trp53* (JAX stock #008462) mouse lines together. Adipocyte-specific p53-knockout can be induced by doxycycline-diet feeding (BioServ, #S3888, Doxycycline 200 mg/kg, sterile, #S4207 control diet, 20.8% protein, 8.7% fat, 2.1% fiber, 3.75 kcal/gm) from weaning until 6–10 weeks old. After 2 weeks doxycycline feeding, the diet was replaced by regular food in animal facility, and proceeded to the experiments as indicated.

## High-fat diet feeding

For dietary alterations, mice were individually housed and fed with a high-fat diet (HFD, 60% kcal in fat, D12492; Research Diets, New Brunswick, NJ), or chow diet.

## Fasting and refeeding experiments

the food was removed at 5pm for 24 hr, followed with or without refeeding for another 24 hr. Tissues were dissected for processing.

## Isolated primary adipocytes

Freshly harvested adipose tissue was digested by collagenase in Krebs-Ringer bicarbonate (KRB) buffer as described in *Ding et al., 2014* and *Liu et al., 2006*.

## Analysis of plasma parameters

Mouse blood was drawn from the tail vein and decanted directly into heparinized capillary tubes (Fisher Scientific). Triglyceride, and free fatty acid levels were measured with standard enzymatic colorimetric assays (Pointe Scientific, Inc and BioVision). Insulin levels were determined by ELISA with kits (Alpco Diagnostics).

## Tissue harvesting

mice were sacrificed, and tissues were immediately frozen in liquid nitrogen and stored at −80°C until biochemical analysis.

## Glucose tolerance test (GTT)

Mice were fasted overnight (12 hr), weighed and administered with 1 g/kg body weight of 20% D-glucose by intraperitoneal cavity injection. Tail vein blood was collected before and at various times (*Figure 3E*) after injection for measurement of glucose concentration (Accu-Chek, Roche).

Animals were maintained in a pathogen-free animal facility at 21°C under a 12 hr light/12 hr dark cycle with access to a chow diet (CD, 2918; Harlan Teklad Global Diet, Madison, WI). All animal studies were performed in accordance with the guidelines and under approval of the Institutional Review Committee for the Animal Care and Use of Boston University (Protocol #201800404).

## Cell culture

3T3-L1 fibroblasts culture and differentiation were previously described in *Liu and Pilch, 2008*. The inducible p53 null 3T3-L1 stable cell lines were generated using lentiviral plasmid based CRISPR/Cas9 genome editing, in which U6 promoter-driven guide RNA, Tet-On 3G doxycycline-inducible promoter-driven cas9 and puromycin selection marker. Lentivirus were packing used 3rd generation packing system. The stable cell lines were obtained after lentivirus transduction and puromycin

selection. For LAL and ATGL silencing, and LAL overexpression, 3T3-L1 adipocytes (day 10 post differentiation) were transfected with control, LAL-directed siRNA, or pcDNA3-LAL using electroporation (*Saito et al., 2007*). 3T3-L1 fibroblasts were commercially available from ZenBio. This is a reliable and widely used resource, and has nearly 100% differentiation into lipid-loaded adipocytes, which confirms the cells' identity. Routine PCR test for mycoplasma contamination was negative all the time.

## Quantitative RT-PCR

Total RNA was isolated from indicated tissues or cells with TRIzol reagent (Invitrogen), and the cDNA was synthesized using Reverse Transcription System (Promega). Real-time PCR was performed with the ViiA7 detection system (Applied Biosystems) using Fast SYBR Green Master Mix (Applied Biosystems). Gene expression levels were normalized to 36B4 and presented relative to the wild type (*Liu and Pilch, 2016*).

## Reagents

Dexamethasone, 3-isobutylmethylxanthine, insulin, sodium fluoride, sodium orthovanadate, fetal bovine serum (Australian origin), benzamidine, and mouse immunoglobulin G (IgG) were purchased from Sigma. LB base, ampicillin, kanamycin, aprotinin, leupeptin, and pepstatin A were obtained from American Bioanalytical (Natick, MA). Calf serum was purchased from Life Science, and Dulbecco's modified Eagle's medium (DMEM) was from Mediatech (Herndon, VA). Transfection reagent and the pcDNA 3.1 expression vector were purchased from Life Science. A BCA protein assay kit was from Pierce. Protein A or G magnetic beads was from Santa Cruz Biotechnology (Santa Cruz, CA). Penicillin, streptomycin, and trypsin were purchased from Life Science. Fatty acid, Lactate, Glucose, ROS, GSSG, and GSH were measured by using commercial available kits.

## Antibodies and western blotting

Monoclonal antibodies recognizing PTRF (2F11), caveolin-1 (7C8), have been previously described (*Vinten et al., 2001*; *Souto et al., 2003*). The following antibodies were commercially acquired: anti-p53 and tubulin were from Cell Signaling; anti-actin was from Sigma; anti-transferrin receptor was from Zymed Laboratories. Primary antibodies were detected in Western blots using secondary antibodies conjugated to horseradish peroxidase (Sigma) diluted 1:3000 and chemiluminescent substrate (PerkinElmer Life Sciences), followed by detection by Fujifilm LAS-4000 Image Analyzer.

## Nucleus fractionation

Nucleus fraction was prepared according to sucrose centrifugation method (Nuclei Isolation Kit, 'Nuclei PURE Prep', Sigma, NUC201). This protocol incorporates centrifugation through a dense sucrose cushion to protect nuclei and strip away cytoplasmic contaminants.

## Co-immunoprecipitation

The whole cell/tissue lysates or sonicated nucleus fraction was solubilized with 1% Triton X-100. Insoluble material was removed by pelleting for 10 min in a microcentrifuge. Indicated antibodies and nonspecific mouse or rabbit IgGs were incubated with the supernatant 1 hr at 4°C, then 20–40 μl of protein A/G magnetic beads was added for 2 hr to overnight. The supernatant with unbound proteins was collected, and the beads were washed four times and eluted with SDS-PAGE loading buffer containing 2% SDS.

## Statistics

All results are presented as mean ± SD. P values were calculated by unpaired Student's t-test. *p<0.05, **p<0.01, and ***p<0.001. p<0.05 was considered significant throughout. For cultured and primary isolated cells, all experiments were performed independently at least three times. Animal studies were from 4 to 6 animals per group.

## Acknowledgements

This work was supported by National Institutes of Health Grant DK-112945 (to LL).

## Additional information

### Funding

| Funder | Grant reference number | Author |
|---|---|---|
| National Institute of Diabetes and Digestive and Kidney Diseases | DK-112945 | Libin Liu |

The funders had no role in study design, data collection and interpretation, or the decision to submit the work for publication.

### Author contributions

Hong Wang, Conceptualization, Data curation, Investigation, Methodology, Project administration; Xueping Wan, Investigation, Methodology; Paul F Pilch, Validation, Investigation, Methodology; Leif W Ellisen, Susan K Fried, Conceptualization, Resources, Formal analysis, Methodology, Writing - review and editing; Libin Liu, Conceptualization, Resources, Data curation, Supervision, Funding acquisition, Validation, Investigation, Methodology, Writing - original draft, Project administration, Writing - review and editing

### Author ORCIDs

Paul F Pilch (ID) http://orcid.org/0000-0003-1997-0499
Libin Liu (ID) https://orcid.org/0000-0001-5056-1517

### Ethics

Animal experimentation: All animal studies were performed in accordance with the guidelines and under approval of the Institutional Review Committee for the Animal Care and Use of Boston University (Protocol #201800404).

### Decision letter and Author response

Decision letter https://doi.org/10.7554/eLife.63665.sa1
Author response https://doi.org/10.7554/eLife.63665.sa2

## Additional files

### Supplementary files

• Transparent reporting form

### Data availability

All data generated or analysed during this study are included in the manuscript and supporting files. Source data files have been provided for figures.

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
