## [Decision Letter]

**Acceptance summary:**

This set of findings reveals an unexpected and important role for the tumor suppressor p53 in adipose tissue function beyond its well-known roles in apoptosis and senescence. The actions of p53 in mediating an AMP Kinase directed regulation of a lysosomal lipase that contributes to hydrolyzing triglycerides to fatty acids for fuel in starvation adds a new dimension to metabolic programming.

**Decision letter after peer review:**

Thank you for submitting your article "An AMPK-dependent, non-canonical p53 pathway plays a key role in adipocyte metabolic reprogramming" for consideration by *eLife*. Your article has been reviewed by two peer reviewers, one of whom is a member of our Board of Reviewing Editors, and the evaluation has been overseen by David James as the Senior Editor. The following individual involved in review of your submission has agreed to reveal their identity: Vishwajeet Puri (Reviewer #2).

The reviewers have discussed the reviews with one another and the Reviewing Editor has drafted this decision to help you prepare a revised submission.

Summary:

This is a very interesting study that breaks new ground related to a new role of p53 in regulating adipocyte metabolism. The experiments are well done, and appear to reveal a new axis of high importance in modulating mitochondrial flux and in turn whole body metabolism. In particular, p53 is shown to modulate the lysosomal acid lipase LAL expression which in turn regulates fatty acid availability for mitochondrial oxidation. Such regulation effects circulating glucose and fatty acid levels in fasting, thus controlling systemic metabolic control.

Essential revisions:

1) Subsection “Inducible adipocyte-specific p53 knockout (p53-iAKO) mice show decreased lipolysis after fasting and improved metabolic phenotypes upon high fat diet feeding”, Figure 7E: This is one of the most interesting data points, in that both blood FA and glucose are decreased in fasting. Is the decreased glucose due to decreased hepatic glucose output in agreement with the concept that fatty acids derived from adipocytes drive hepatic gluconeogenesis? Or increased glucose utilization in muscle? Have clamps been done to check on this hypothesis? Related to this issue of systemic metabolism, the p53-KO cells have more lactate production. What happens to the lactate in the in-vivo mouse model? Do they show elevated lactate in their circulation? Does it cause lactic acidosis in animals? The authors could provide at least a better picture of the overall in vivo metabolic profile in the KO mice?

2) In Figure 4, the authors claim that the change in lipolysis was not due to the traditional neutral lipolysis pathway, although knocking down LAL alone is not sufficient to suppress ISO stimulated lipolysis. Comparing blue bars in Figure 4D (control vs. ISO) shows that there is a significant increase in lipolysis that is independent of LAL/p53. Did the authors look at ATGL expression? The simultaneous knockdown of ATGL and LAL followed by induced lipolysis analysis would reveal the role of other lipases. Please show the total and phosphor-HSL and PLIN western blots (might be supplementary data?).

3) It is crucial to show the levels of phospho-p53 (instead of total), and p-AMPK in Figure 7B and C in the fat pads of the -/+ dox treated animals to support the conclusions. Western blots of phospho-p53 and phospho-AMPK are also crucial for interpreting the results among control, fasted, and refed animals and its correlation, as shown in Figure 7.

[Editors' note: further revisions were suggested prior to acceptance, as described below.]

Thank you for resubmitting your work entitled "An AMPK-dependent, non-canonical p53 pathway plays a key role in adipocyte metabolic reprogramming" for further consideration by *eLife*. Your revised article has been evaluated by David James as the Senior Editor and a Reviewing Editor.

The manuscript has been improved but there is one remaining issue that you may want to address before acceptance, as outlined below:

In your model figure you have "Nutrient Availability" leading to (arrow) AMPK, but in fact your hypothesis is that decreased nutrient availability is what causes AMPK activation. You may want to clarify this apparent inconsistency, since the model will get the attention of the readers and should be clear. We will leave it to your decision if you wish to change this model cartoon or not. Once you have made that decision, we will be able to accept the manuscript for publication.

---

## [Author Response]

Essential revisions:1) Subsection “Inducible adipocyte-specific p53 knockout (p53-iAKO) mice show decreased lipolysis after fasting and improved metabolic phenotypes upon high fat diet feeding”, Figure 7E: This is one of the most interesting data points, in that both blood FA and glucose are decreased in fasting. Is the decreased glucose due to decreased hepatic glucose output in agreement with the concept that fatty acids derived from adipocytes drive hepatic gluconeogenesis? Or increased glucose utilization in muscle? Have clamps been done to check on this hypothesis? Related to this issue of systemic metabolism, the p53-KO cells have more lactate production. What happens to the lactate in the in-vivo mouse model? Do they show elevated lactate in their circulation? Does it cause lactic acidosis in animals? The authors could provide at least a better picture of the overall in vivo metabolic profile in the KO mice?

We performed additional experiments, adding 11 new data panels in Figure 7, providing a better overall metabolic profile. Specifically, we measured hepatic gluconeogenesis by using lactate tolerance test and glucose utilization by in vivo glucose uptake assay. We conclude the main cause of decreased glucose and FA levels is the glucose utilization, probably due to downregulated lipolysis in adipose tissue. We provide new data supporting the level of lactate was not up-regulated in p53-KO mice, and blood pH did not change either.

2) In Figure 4, the authors claim that the change in lipolysis was not due to the traditional neutral lipolysis pathway, although knocking down LAL alone is not sufficient to suppress ISO stimulated lipolysis. Comparing blue bars in Figure 4D (control vs. ISO) shows that there is a significant increase in lipolysis that is independent of LAL/p53. Did the authors look at ATGL expression? The simultaneous knockdown of ATGL and LAL followed by induced lipolysis analysis would reveal the role of other lipases. Please show the total and phosphor-HSL and PLIN western blots (might be supplementary data?).

Thanks for the reviewer for this key point. We measured ATGL expression levels and performed additional ATGL RNAi knockdown and double knockdown (LAL and ATGL) in 3T3-L1 adipocytes. It seems double knockdown significantly abolished lipolysis, but it was not as low as basal control. Therefore, we cannot completely rule out the potential possibility of other lipase that might be involved. Nevertheless, these data clearly demonstrate LAL-mediated lipolysis plays a key role in lipolysis. The detailed relationships among LAL and other lipases are interesting for future studies.

New total and phosphor-HSL and PLIN western blots have been added in Figure 4—figure supplement 1.

3) It is crucial to show the levels of phospho-p53 (instead of total), and p-AMPK in Figure 7B and C in the fat pads of the -/+ dox treated animals to support the conclusions. Western blots of phospho-p53 and phospho-AMPK are also crucial for interpreting the results among control, fasted, and refed animals and its correlation, as shown in Figure 7.

New data for p53 and AMPK phosphorylation are added (Figure 7Q), which are consistent with our in vitro data. It needs to mention we demonstrate AMPK is the upstream signal pathway for metabolic p53 activation, however it is unclear how p53 affects AMPK pathway in a feedback manner. Our new data show p-AMPK was slightly lower in p53-KO mice, but not significant. Previously a number of publications reported some key components in AMPK pathway were the transcriptional targets of p53. Future studies are needed to rule out the relationship between these two pathways in adipocyte metabolism.

[Editors' note: further revisions were suggested prior to acceptance, as described below.]

The manuscript has been improved but there is one remaining issue that you may want to address before acceptance, as outlined below:In your model figure you have "Nutrient Availability" leading to (arrow) AMPK, but in fact your hypothesis is that decreased nutrient availability is what causes AMPK activation. You may want to clarify this apparent inconsistency, since the model will get the attention of the readers and should be clear. We will leave it to your decision if you wish to change this model cartoon or not. Once you have made that decision, we will be able to accept the manuscript for publication.

We thank reviewers for the comment on the model figure, and have changed “Nutrient Availability” to “Nutrient starvation”, which causes AMPK activation. This change has been updated in all files.